# Mitochondria Localized microRNAs: An Unexplored miRNA Niche in Alzheimer’s Disease and Aging

**DOI:** 10.3390/cells12050742

**Published:** 2023-02-25

**Authors:** Jazmin Rivera, Laxman Gangwani, Subodh Kumar

**Affiliations:** 1Center of Emphasis in Neuroscience, Department of Molecular and Translational Medicine, Paul L. Foster School of Medicine, Texas Tech University Health Sciences Center, El Paso, TX 79905, USA; 2Department of Veterinary Pathobiology and Bond Life Sciences Center, University of Missouri, 1201 Rollins Street, Columbia, MO 65211, USA; 3L. Frederick Francis Graduate School of Biomedical Sciences, Texas Tech University Health Sciences Center El Paso, El Paso, TX 79905, USA

**Keywords:** mitochondrial miRNAs, Alzheimer’s disease, mitochondrial dysfunction, synaptic energy, aging

## Abstract

Mitochondria play several vital roles in the brain cells, especially in neurons to provide synaptic energy (ATP), Ca^2+^ homeostasis, Reactive Oxygen Species (ROS) production, apoptosis, mitophagy, axonal transport and neurotransmission. Mitochondrial dysfunction is a well-established phenomenon in the pathophysiology of many neurological diseases, including Alzheimer’s disease (AD). Amyloid-beta (Aβ) and Phosphorylated tau (p-tau) proteins cause the severe mitochondrial defects in AD. A newly discovered cellular niche of microRNAs (miRNAs), so-called mitochondrial-miRNAs (mito-miRs), has recently been explored in mitochondrial functions, cellular processes and in a few human diseases. The mitochondria localized miRNAs regulate local mitochondrial genes expression and are significantly involved in the modulation of mitochondrial proteins, and thereby in controlling mitochondrial function. Thus, mitochondrial miRNAs are crucial to maintaining mitochondrial integrity and for normal mitochondrial homeostasis. Mitochondrial dysfunction is well established in AD pathogenesis, but unfortunately mitochondria miRNAs and their precise roles have not yet been investigated in AD. Therefore, an urgent need exists to examine and decipher the critical roles of mitochondrial miRNAs in AD and in the aging process. The current perspective sheds light on the latest insights and future research directions on investigating the contribution of mitochondrial miRNAs in AD and aging.

## 1. Alzheimer’s Disease

Alzheimer’s disease (AD) is a progressive neurological disorder that affects approximately 50 million people worldwide [1]. Alzheimer’s disease is ranked as the seventh leading cause of death in the United States and is most associated with dementia among older adults (https://www.nia.nih.gov/health/alzheimers-disease-fact-sheet, accessed on 15 October 2022). Dementia refers to a loss of cognitive function, with a broad range of symptoms depending on the stage of diagnosis. Memory loss, an inability to carry out daily activities and a difficulty in organizing thoughts and thinking logically are symptoms associated with brain atrophy caused by AD.

Researchers continue to unravel complex changes involving the AD brain. Alzheimer’s has been divided into familial and sporadic forms associated with the stages of the disease. According to studies, early-onset (familial) cases account for ~10% of all patients with AD, primarily affected individuals below 65 years of age. On the other hand, late-onset (sporadic) contributes up to 30% of all patients with AD developing before the age of 65 [2]. Certain AD cases are caused by inherited changes within genes. Early-onset AD has been linked to mutations on chromosome 19, entailing the apolipoprotein E (APOE) gene. However, a large body of research suggests that other genetic components may also be involved in the manifestation of AD, such as mutations in amyloid-beta precursor protein (AβPP), presenilin1, and presenilin 2 genes attributing to the overproduction of Aβ plaques. A group of proteins and peptides including transthyretin, calcitonin, gelsolin, amylin, atrial natriuretic peptide and amyloid-beta have been hallmarked as the fibrillar components of diseases that are characterized by amyloid deposits [2].

We recently conducted a meta-analysis study on the deregulated mitochondrial microRNAs (miRNAs) in AD [3]. We reviewed and proposed the potential roles of mitochondrial miRNAs in mitochondrial activities and synapse function. However, our meta-analysis study lacked information about brain mitochondria localized miRNAs and their relevance in AD. The main objective of the current article is to unveil the mitochondria localized and associated miRNAs and their critical role in maintaining the normal mitochondrial function and in AD pathogenesis.

## 2. Mitochondria and Alzheimer’s Disease

The mitochondrion is an important cellular component known for its role in bioenergetics, metabolism, signaling pathways and cell viability [4]. Mitochondria play several vital roles in the brain cells, especially in neurons to provide synaptic energy (ATP), Ca^2+^ handling, reactive oxygen species (ROS) production, apoptosis, mitophagy, axonal transport and neurotransmission [5,6,7]. It is established that a healthy pool of mitochondria provides necessary energy to the neurons for proper neuronal function and synaptic activity. The mitochondria also protect neurons from oxidative stress and free radical damages [5,6,7].

It is reported that mitochondrial dysfunction is a new hallmark of AD initiation and progression [8,9,10]. A number of mitochondrial abnormalities are reported in the AD brain, including disrupted mitochondrial bioenergetics, increased oxidative stress, mitochondrial genomic stress, abnormal mitochondrial fusion and fission, mitochondrial axonal trafficking deficits and abnormal mitochondrial distribution, impaired mitochondrial biogenesis, abnormal endoplasmic reticulum–mitochondrial interaction, impaired mitophagy and impaired mitochondrial proteostasis [5,6,7,8]. All these mitochondrial dysfunctions are caused by multiple biological, genetic, and environmental factors, including Aging, Aβ and p-tau toxicities, inflammation, miRNAs deregulation, gender differences and environmental toxins [3,8,9,10,11]. Mitochondrial dysfunction has a significant negative impact on synaptic activities in AD, such as impaired calcium signaling, reduced synaptic energy, defective neurotransmission, and synaptic dysfunction [5,6,7,8,9,10,11]. As mitochondrial genome disturbance is one of the contributing factors for mitochondrial dysfunction in AD, it is important to uncover the mitochondrial genome-associated miRNAs deregulation and their impact on mitochondrial and synaptic dysfunction in AD.

## 3. MicroRNAs

MicroRNAs (miRNAs) are regulators of human genes, and their therapeutic relevance has been explored in human diseases, including AD [3]. MiRNAs are present throughout the cells and some miRNAs are localized to cellular organelles. Subcellular compartmentalization and localization of miRNAs, miRNA-induced silencing complex (miRISC) and target mRNA have been observed to localize in multiple subcellular compartments including mitochondria, nucleus, rough endoplasmic reticulum, processing (P)-bodies, early/late endosomes, multivesicular bodies, lysosomes and synaptosomes [3,12,13,14,15,16,17]. MiRNAs are small and noncoding RNAs that regulate gene expression through the process of binding to messenger RNA (mRNA) [3,16,17,18].

According to studies, miRNAs are attributed to the post-transcriptional regulation of mitochondrial gene expression and control mitochondrial activities [19]. MiRNAs are transcribed as double-stranded RNA, also known as pre-miRNA. On the other hand, mature miRNAs bind to Argonaute proteins that form the RNA-induced silencing complex ribonucleoprotein (RISC) [4]. Pre-miRNAs and mature miRNAs found in the mitochondria interfere with mitochondrial genome derived mRNA, which affects the mitochondria and cell function. Through complementary base pairing RISC binds to 3’-UTR, thus downregulating gene expression. The result of this process initiates mRNA degradation or translational repression affecting the production of protein levels [3,20]. MiRNAs found in the central nervous system play a role in the translation and degradation of genes. According to recent research, synaptic activity and function has been linked to miRNAs due to their ability to interact with mRNAs, resulting in physiological changes [16].

## 4. MiRNAs Localization in Mitochondria

Specific mitochondrial miRNAs are miRNAs that are localized in the mitochondria. The presence of miRNA in mitochondria has only been discovered in the past decade in living organisms. Several studies have identified the presence of miRNAs in the mitochondria and their important roles in local mitochondrial protein synthesis, and in the regulation of mitochondrial functions [12,13,14,17]. Initially, in 2009, Kren et al., identified the rat-liver-derived mitochondrial miRNAs and unveiled their roles in apoptosis [12]. The five miRNAs—miR-130a, miR-130b, miR-140, miR-320 and miR-494—were identified as mitochondrial enriched miRNAs, and most of them are involved in the expression of genes associated with apoptosis, cell proliferation, and differentiation [12].

In 2011, Barrey et al. studied the miRNA localized in the mitochondria isolated from human skeletal primary muscular cells [17]. The three precursor miRNAs—pre-mir-302a, pre-let-7b and mir-365—were found to be localized in the mitochondria of human myoblasts [17]. In the same year, Bendiera et al. studied mitochondrial-enriched miRNAs in HeLa cells [14]. First, the author separated the mitochondrial and cytosolic fraction from the same samples of HeLa cells. Next, they performed the miRNAs analysis in both mitochondria and cytosolic fraction. A total of 57 miRNAs was found to be significantly deregulated in the mitochondria versus the cytosol. The three nuclear-encoded miRNAs signatures hsa-miR-494, hsa-miR-1275 and hsa-miR-1974 were found to be consistently upregulated in mitochondria [14]. Further, the author found the presence of Argonaute 2 protein in the mitochondria, which suggests that mitochondria miRNAs could modulate the expression of local mitochondrial proteins [14].

Therefore, the precise roles and therapeutic relevance of these mitochondria localized miRNAs are still unknown. Hence, despite all information and available research [3,20,21,22,23,24,25], mitochondrial miRNAs are still an unexplored niche in human diseases [22], neurological disorders.

## 5. Mitochondria Localized miRNAs and Mitochondrial Dysfunction in AD

Alzheimer’s disease is tied to mitochondrial dysfunction, including overproduction of ROS, low ATP production and influx of calcium ion. The disruption of calcium homeostasis causes damage to the mitochondria, resulting in damage to synaptic dysfunction. Together, all these features contribute to the dysfunction of the mitochondria, leading to the progression of AD [2]. Furthermore, Aβ plaques and NFTs have been associated with impeding mitochondrial function increasing neuronal deficits, leading to neurodegeneration associated with neurological disorders such as AD [26].

Mitochondrial respiration defects are the characteristics found in the brains of patient with AD. Decreased neurological function, impairments and brain atrophy have been tied to the reduction in mitochondrial enzyme activity in the premorbid cognitive level [18]. MiR-338 is a brain-specific miRNA that has been shown to modulate the expression of cytochrome c oxidase IV (COXIV), a protein within the ETC which contributes to ATP production, in neuronal cells [27]. Moreover, the expression of miR-338 is correlated to the reduction in COXIV mRNA and reduction in protein levels. A study based on the control of mitochondrial activity by miRNAs revealed overexpression of miR-338 reduced mitochondrial oxygen consumption, metabolic activity, and ATP production [21].

MiRNAs mediated mitochondrial impairment decreases ATP production, alters calcium influx, and increases ROS production. A decrease in ROS concentration is essential for normal cell signaling; on the other hand, a high concentration of ROS damages macromolecules and increases the mutation rate of mitochondrial DNA (mtDNA). Studies have linked mitochondrial dysfunction with age and oxidative stress, contributing to neurodegenerative diseases [18].

Furthermore, quite a few specific miRNAs play important roles in mitochondrial function, as well as in various aspects of synaptic plasticity, including synaptotoxicity, synaptic activity and neurotransmission [28]. The miR-132, miR-34a, miR-484, miR-218, miR-455-3p, miR-34a and miR-212 are the potential miRNAs that were studied in mitochondria mediated synaptic functions (Table 1) [29]. For example, miR-132 downregulated in AD and involved in the regulation of PTEN, FOXO3a, P300, NOS1 and MMP- 9 genes thereby enhances neurotransmission and synaptic plasticity [30,31]. MiR-34a upregulated in AD causes synaptic plasticity dysfunction via the modulation of VAMP2, SYT1, HCN, NR2A and GLUR1 proteins [32].

Another study in 2019, Qian et al., provided evidence that the overexpression of miR-338 in mice is associated with neuropathology in AD. The results from in vitro cultured neurons showed an increase in NF-kB activity due to the regulation of miR-338 that might contribute to inflammatory states in patients with AD. Analysis of the lysates of hippocampal tissue from 6-month-old 5XFAD transgenic (TG) mice demonstrated that the transcription of miR-338 promoted the expression of BACE1, leading to an increase in Amyloid beta formation resulting in neuroinflammation and cognitive dysfunction [33].

Previous studies identified several miRNAs deregulated in AD brain, blood, serum, plasma, CSF and AD mouse model [3,15,16]. MiRNAs deregulation linked with AD in two ways; (i) the deregulation of miRNAs could be initiated by AD pathogenic factors such as Aβ, p-tau, inflammation, aging, oxidative stress, and mitochondrial DNA damage, and/or (ii) the genetic alteration of miRNAs could be a contributing factor in AD progression. In both ways, miRNAs significantly contribute to AD via the modulation of expression of disease associated genes/proteins and the regulation of cellular pathways either in a positive or negative way. Since normal mitochondrial function is crucial to control AD, we recently summarized the potential miRNAs; those are deregulated in AD and involved in several aspects of mitochondrial function such as mitochondrial biogenesis, dynamics, mitophagy, ATP production, oxidative stress and apoptosis, that ultimately lead to impaired synaptic function in AD [3]. Based on the miRNA’s location, association and pivotal roles, we categorized them as ‘mitochondrial localized miRNAs’ and ‘mitochondria associated miRNAs’ (Table 1). Mitochondria localized miRNAs are supposed to be present and expressed within mitochondria and regulate mitochondrial functions. While mitochondria associated miRNAs could be some common miRNAs, those significantly modulate mitochondrial function. Therefore, several miRNAs are identified in AD, those regulate the key mitochondrial functions; however, it is unclear if these miRNAs are expressed within the mitochondria and transcribed from mitochondrial genome, or what their impact is on the levels of local mitochondrial proteins. Important mitochondria localized miRNAs and mitochondria-associated miRNAs, their location and their cellular functions in human diseases are summarized in Table 1.

**Table 1 cells-12-00742-t001:** Mitochondria localized miRNAs and mitochondria-associated miRNAs, their location and cellular function in AD and other diseases.

**Mitochondria Enriched/Localized miRNAs**						
**miRNAs**	**Location**	**Function**	**Target**	**Disease**	**Other Cellular Function**	**Reference**
miR-130a	Mitochondria	Induced neurotoxicity and cell apoptosis	DAPK I	AD and PD	Regulate neurotransmitter synthesis	[12]
miR-130b	Mitochondria	Inhibits cell proliferation and induces cell apoptosis	PTEN and AKT	AD and PD	Regulators of lipid homeostasis and lipoprotein trafficking	[12]
miR-140	Mitochondria	Inhibits mitochondria dysfunction and enhances autophagy	PINK I, ADAM 10 and SOX2	AD, stroke and PD	Inhibits cell proliferation	[12]
miR-494	Mitochondria	Affects ATP synthesis and leads to mitochondrial dysfunction	DJ-1	AD and PD	Exacerbate oxidative stress-induced neuronal damage by reducing DJ-1 expression	[12]
miR-132	Mitochondria	Inhibits complex 1 of mitochondria and reduces aerobic respiration	PTEN, FOXO3a, P300, NOS1	AD, HD	Decreased AGO2 function	[29]
miR-34a	Mitochondria	Synaptic plasticity dysfunction	VAMP2, SYT1, HCN, NR2A, GLUR1	AD, HD and PD	Downregulates antioxidant mitochondrial enzymes	[32]
miR-365	Mitochondria	Accumulation of ECM components and secretion of inflammatory cytokines	BDNF and p-TrkB	AD and fibrosis	Inhibits apoptosis	[20]
miR-338	Mitochondria	Promotes the regulation of ROS generation	COXIV and ATP5G1	Sclerosis and AD	Axonal outgrowth	[21]
miR-484	Mitochondria	Inhibits mitochondrial fission	Fis1, BCL2L13	PD	Inhibits apoptosis	[29]
miR-218	Mitochondria	Inhibits mitophagy and mitochondrial clearance	PRKN	AD, PD and HD	Protects against neurotoxicity that is caused by metallic ions and other neurotoxins	[29]
miR-212	Mitochondria	Enhance neurotransmission and involved in synaptic plasticity	PTEN, FOXO3a, P300, NOS1	AD	Inhibits apoptosis	[29]
miR-146a	Mitochondria	Induces inflammatory response and contributes to cell aging	IRAK-1, TRAF-6	AD	Inhibits apoptosis	[4]
**Mitochondria Associated miRNAs**						
**miRNAs**	**Location**	**Function**		**Target**	**Disease**	**Reference**
miR-705	Cytoplasm	Mitochondrial dysfunction		WARS	Osteoporosis and AD	[13]
miR-494	Cytoplasm	Affects ATP synthesis and leads to mitochondrial dysfunction		HLF, Gli3, Mal, LIF, LCOR, Tfam	AD and PD	[13]
miR-202-5p	Cytoplasm	Synaptic plasticity dysfunction		BCL2, LBR, CD28, HLF	AD and PD	[13]
miR-181c-5p	Cytoplasm	Imbalance of ROS generation and causes mitochondria dysfunction		COX1	AD	[34]
miR-155	Cytoplasm	Regulation of inflammatory responses		CD1d	AD	[35]
miR-223	Cytoplasm	Regulation of inflammatory and immune responses		MSMO1, HMGCR, SR-B1	AD, PD, ALS	[35]

## 6. Aging and Synaptic Dysfunction in Alzheimer’s Disease

Aging is a known factor that increases the progression of brain deterioration, causing epigenetic changes, protein damage and mitochondrial dysfunction, thus contributing to AD progression. As an individual begins to age the production of mitochondrial reactive oxygen species (ROS) is instigated, which alters the electron transport chain. The result of the disruption to the electron transport chain has been linked to apoptotic cell death. According to a research article published in 2021, mutations found in APP, PS1 and PS2 are associated with early onset-AD, whereas age related factors such as ROS production, mitochondrial DNA changes and epigenetic factors have been observed in sporadic AD [8]. Cognitive function depends on synaptic activity and ATP production. Elderly individuals with AD experience synaptic mitochondria interference through the accumulation of Aβ and p-tau proteins. The accumulation of these proteins at the nerve terminals and synapses results in defective and inactive mitochondria affecting the communication of neuronal cells.

Recent evidence from postmortem brains of animals and clinical studies suggest that mitochondria play crucial roles in aging and neurodegenerative diseases [8]. Mitochondrial dysfunction was noticed in the postmortem brains of neurodegenerative disease expressing mutant proteins such as Aβ, mutant Htt, mutant SOD1 and mutant DJ1, among others. Furthermore, the abnormal expression of mitochondrial encoded genes has been observed in AD brains, suggesting that mitochondrial metabolism plays a crucial role in AD.

## 7. Mitochondrial miRNAs and Synapse Dysfunction in Alzheimer’s Disease

Synapses are crucial for neuronal communication and cognitive function. Both chemical and electrical synapses compose the complexity of the synaptic network in the human brain. Chemical synapses receive signals through various presynaptic neurons to a postsynaptic neuron. Electrical synapses, on the other hand, form connections through gap junctions that result in the direct transfer of ions.

Neuronal function and plasticity are correlated to the fluctuance of calcium ion concentration. Calcium ions result in the depolarization of neurons affecting synaptic activity. Calcium channels are triggered upon transport of calcium ions into the presynaptic terminal, which releases neurotransmitters through exocytosis. Mitochondria work to maintain homeostasis by regulating the concentration of calcium depending on ATP consumption. A high concentration of calcium triggers the activation of mitochondrial permeability transition pores (mPTPs), resulting in apoptosis [36].

Synaptic mitochondria play an essential role in synaptic activity through the fluctuation in levels of calcium ions. Synaptic stress is a known pathological hallmark for AD. As another example, Aβ and Aβ-associated cellular changes cause neuronal perturbations and synaptic changes in AD. Aβ in young AD mouse models displayed extracellular deposition. Synaptic mitochondria displayed an increase in Aβ levels and changes were noticed in the function of non-synaptic mitochondria in AD models. This suggests that mitochondria are more susceptible to Aβ damage and mitochondrial stress, causing symptoms associated with AD [37]. A recent study showed the initiation of apoptosis through the activation of caspase-3 in the hippocampal dendritic spines of mice models leading to early synaptic dysfunction and dendritic spine loss [37].

Synaptic activity is modulated by axonal transport, which is dependent on mitochondrial density [37]. Presynaptic sites are known to influence synaptic vesicle release, impairing axonal transport, and contributing to the pathogenesis of AD. Patients with AD were demonstrated to have axonal degeneration that accumulates in the mitochondria [37].

During a 2020 study, Naval Medical Research Institute (NMRI) mice were monitored during the aging process. Several changes in cognitive performance and mitochondrial metabolism were observed over a span of 24 months [38]. Additionally, studies show that impaired axonal transport reduces synaptic mitochondrial density and interferes with mitochondrial trafficking through the AD synapse. Together, altered axonal transport and fluctuations in calcium ions concentration cause impaired synaptic vesicle release and synaptic distress associated with AD pathogenesis. The modulation of mitochondrial and synaptic proteins via miRNAs is critical for the normal synapse function and very important to understand the pathophysiology of plasticity-related diseases. Since the synaptic activities are closely tied with healthy mitochondrial function and a consistent supply of ATP at synapse, therefore each mitochondrial component (here miRNAs) needs in-depth investigation.

## 8. Mitochondria-Associated miRNAs and Mitochondrial Dysfunction in AD

Mitochondrial miRNAs paved the way to further understand molecular mechanisms of translocation of miRNAs from the nucleus to the mitochondria. By mapping the nuclear genome of mitochondrial miRNAs, studies revealed a link between mitochondrial function and disease. Studies have shown how mitochondrial miRNAs can target mitochondrial genome and can harbor sequences [19]. The targeting of the mitochondrial genome could directly influence the energetic, oxidative, and inflammatory status of cells, which may cause changes in an organism. Through the effect of ATP synthesis, mitochondrial miRNAs can influence mitochondrial function. MiR-181c-5p is a known target of mitochondrial miRNAs that originates from the nuclear genome and translates to mitochondria. The over expression of miR-181c-5p causes the loss of mt-COX1 protein and results in an imbalance in ROS generation. Recently, a study of miR-181c-5p in rats showed that altered mitochondrial metabolism and ROS generation causes heart failure in animals [34].

In 2010, Bian et al. identified the mouse liver mitochondria-associated miRNAs and studied their potential biological functions [13]. A set of three miRNAs—miR-705, miR-494 and miR-202-5p—were identified as potential mitochondria-associated miRNAs [13]. These miRNAs have several putative targets related to mitochondria-specific functions, such as tryptophanyl-tRNA synthetase and transcription factor A, and may be involved in the modulation of mitochondria and cell-specific functions [13]. In this study, eight-week-old mice were treated with STZ (150 mg/kg) and harvested 14 days post-injection, and mitochondria were isolated from the liver. A Western blot analysis using antibodies against cytochrome c and AGO2 demonstrated the purity of liver mitochondria. These results suggested that mitochondria associated miRNAs were involved in mitochondrial dysfunction, causing progression in neurodegenerative diseases such as Alzheimer’s.

## 9. Other Cellular miRNAs and Regulation of Mitochondrial Function

The metabolic regulation of mitochondria is regulated by peroxisome proliferator-activated receptor y coactivator 1 (PGC-1), which interacts with many other transcriptional factors. Located in the first intron of the PGC-1 gene is the miR-378. Studies involving miR-378 in mice, which is controlled by PGC-1b, regulate mitochondrial metabolism and the homeostasis of the organism. Previous studies have found a link between miR-378’s targets and metabolic protein repression. However, several other miRNAs are involved in metabolic homeostasis based on mice studies. For example, miR-33 is an element binding protein gene that has been associated with cholesterol levels and lipid homeostasis by targeting adenosine triphosphate through the binding of cassette transporter A1. Other miRNAs have been linked to the regulation of glucose metabolism involving the silencing of miR-103/107, affecting glucose homeostasis and insulin sensitivity [39].

## 10. Mitochondria Associated miRNAs and Central Nervous System Function 

Deregulation of the mitochondria has been implicated in the onset and the progression of neurological disorders such as Alzheimer’s, Parkinson’s and Huntington diseases [40]. An increase in ROS generation is seen as people age and adds to the oxidative damage seen during mitochondrial respiration. Mitochondrial miRNAs are known for disrupting the respiratory chain complexes and increasing ROS production, resulting in mitochondrial damage.

In 2014, Rippo et al. focused on investigating miRNAs expression in HUVEC cells, where miR-146a was compared against younger cells and regulated with specificity to Bcl-2. The results were examined using different cell models in *in vivo* which demonstrate the complexity of aging process [40]. For example, miR-155 and miR-146a are associated with inflammation in the nervous system by the activation of TLR7. The dysregulation of mitochondrial miRNAs affects the mitochondria causing immune responses in the brain. Mitochondrial damage and cell stress starts promoting inflammation at neuronal synapses that spread across the postsynaptic membrane to neighboring neurons.

In 2015, Wang et al. provided evidence linking mitochondria-associated miRNA expression in relation to controlled cortical impact (CCI) injury in rats. The experiment confirmed that several mitochondrial-associated miRNAs such as miR-155 and miR-223 were elevated in rats subjected to TBI (traumatic brain injury) compared to uninjured rats. It was concluded that mitochondrial-associated miRNAs are crucial to regulating the response to TBI [41].

Another experiment conducted by Wang et al. presented new findings indicating that mitochondria-associated endoplasmic reticulum membranes (MAMs) played a role in neurodegenerative diseases. Analysis was made using subcellular fractionation and TaqMan RT-qPCR to quantify miRNA levels using rat and human brain samples. The results showed evidence of miR-223 causing inflammatory and immune response contributing to neurodegenerative diseases such as AD [42].

In essence, several mitochondrial miRNAs have been associated with controlling mitochondrial function by targeting and affecting different protein expressions. Their modulation, along with the changes in the mitochondria, induces the inflammatory response tied to age-related diseases. Moreover, the early detection and reduction of mitochondrial loss may slow/prevent the progression of neurodegenerative diseases such as Alzheimer’s [4].

## 11. Summary and Future Directions

To summarize, miRNAs are involved in cellular changes associated with neurodegenerative diseases and aging. Alterations in mitochondrial miRNAs expression continue to be an important topic of current research. Identifying mitochondrial miRNAs changes could help to understand the details of mitochondrial dysfunction in AD progression, could forward the invention of mitochondria based diagnostic tools and could allow the development of suitable preventive strategies against Alzheimer’s disease. Current studies have suggested that miRNAs significantly contribute to the development and progression of AD in either positive or negative ways. For this reason, investigating the role of mitochondrial miRNA and understanding their deregulation is necessary to better understand the mitochondrial dysfunction in AD. Several key questions are still unanswered in terms of mitochondrial miRNAs and AD pathogenesis: (1) Are mitochondrial genome encoded miRNAs levels altered in AD? (2) If yes, what is the impact of altered mitochondrial miRNA levels on mitochondrial function? (3) Are any mitochondrial miRNAs altered in response to Aβ and p-tau induced toxicities in AD? (4) Is mitochondrial miRNA deregulation responsible for deprived mitochondrial functions and synaptic activity in AD? and (5) What could be the possible miRNAs-based research strategy to improve the mitochondrial function and synaptic activity in AD? Therefore, further research is needed to evaluate the critical roles that mitochondrial miRNAs might play in mitochondrial and synaptic function in AD. Multi-Omics analysis of brain mitochondria is the best way to understand the genomic and proteomic changes in the mitochondrial genome in AD versus cognitively healthy control brains. The transcriptomic analysis of brain mitochondrial mRNAs, miRNAs, small RNAs and circular RNAs, and the proteomic analysis of mitochondrial proteins within the same sample, will unveil the local RNA-protein interaction in mitochondria. The multi-Omics analysis will provide insight into the potential mitochondrial Omics targets altered in the AD brain. Therefore, a high throughput multi-Omics analysis of the mitochondrial genome and proteome is needed to understand mitochondria-based synaptic dysfunction in AD. Further, a deeper understanding of mitochondrial miRNAs could help to develop mitochondrial biomarkers that could be used as a diagnostic tool to detect early stages of AD. Further, mitochondrial miRNAs research will help to understand mitochondria-based disease pathobiology and to develop novel therapeutic strategies against AD.

## Data Availability

Not applicable.

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
