# Peer review of "Mitochondria Localized microRNAs: An Unexplored miRNA Niche in Alzheimer’s Disease and Aging"

_cells, 2023, doi:10.3390/cells12050742_

Round 1

Reviewer 1 Report

The title and the abstract indicated that this manuscript would focus on discussing the potential role of mitochondria-localized miRNAs in AD and aging. However, the content of the manuscript is stuffed much of other topics rather than focusing on mitomiRs and fails to deliver the perspectives on brain or AD relevant mitomiRs. Several major issues are detailed below:

1. Too much of other subjects. The reviewer agree that it is necessary to set up important background surrounding the main topic. However, this manuscript over-presented unnecessary background topic and lost the main direction which, the authors indicated should be mitomiRs.  

2. mitomiR is a general term for those miRNAs that either translocated into mitochondria or associated with/enriched in mitochondria. The location of mitomiR could have different mechanistic functions in the cell. Authors should clarify this important aspect of mitomiR when they discuss the relevant function of the mitomiR.  

3. Mixed mitomiR and other miRNA concepts: per commonly agreed, mitomiRs are those miRNAs that reside in mitochondria (including translocated or associated/enriched in mitochondria) and function to regulate gene expressions including genes that are important to mitochondria function; other miRNAs are those that do not physically reside on mitochondria but regulate genes involve in mitochondria function. Following authors’ manuscript title and abstract, mitomiRs should be the focus of the discussion. Unfortunately, only a small portion of the manuscript content is relevant to these miRNAs. Many of mentioned miRNAs have no proven to be resided in mitochondria as ‘mitomiRs’. Authors should clarify if they categorized these miRNAs differently than the commonly agreed.

4. Even though this is supposed to be a perspective article, the hypothesis/perspective still should be proposed based primarily on scientific findings. The authors intended to discuss the role of mitomiRs in AD/aging, however, no mitomiR was mentioned to be experimentally confirmed as CNS mitomiR in this manuscript. 

5. The future direction marked several questions but no actual perspective or direction is given.   

6. Line 150: “Based on these reports, it is confirmed that mitochondria retain local miRNAs synthesis mechanism, some potential miRNAs and their precursors.” This is an inaccurate statement. Argonaut protein complexes are mainly responsible for miRNA executive function. The existence of AGO complex in mitochondria suggests that miRNAs found in mitochondria maybe functional but do not confirm that miRNAs are synthesized in mitochondria.

7. Line 149: ‘…and identified human miRNAs that we termed ‘mito-miRs’.’ Bendiera et al (2011) already named the subset of miRNA as ‘mitomiRs’. I don’t see any point to rename these miRNA as ‘mito-miRs’

8. The authors are encouraged to directly utilize original research articles in proposing putative function or mechanism of mitomiRs.

9. The following sentence is identical to a publication. The reviewer suggest the authors modifying the sentence.

Authors’ manuscript Line 147: ‘…provides the first comprehensive view of the localization of RNA interference components to the mitochondria’

Bendiera et al (2011) Abstract: “This study provides the first comprehensive view of the localization of RNA interference components to the mitochondria”

10. Figure 1 did not add any useful information to the manuscript.

11. The manuscript is hard to read and largely unpolished. For example, line 84 ‘MiRNAs are the potential regulators of human genes’—miRNAs are proven regulators of human genes! the acronym ‘miRNAs’ or ‘miRNA’ already appear before the line 93 ‘MicroRNAs (miRNAs) are small..’  and miRNA, microRNA had been used randomly throughout the text. There are many more such minor editing issues in the manuscript.

Author Response

Reviewer 1- Comments and Suggestions for Authors

The title and the abstract indicated that this manuscript would focus on discussing the potential role of mitochondria-localized miRNAs in AD and aging. However, the content of the manuscript is stuffed much of other topics rather than focusing on mitomiRs and fails to deliver the perspectives on brain or AD relevant mitomiRs. Several major issues are detailed below:

Comment 1. Too much of other subjects. The reviewer agrees that it is necessary to set up important background surrounding the main topic. However, this manuscript over-presented unnecessary background topic and lost the main direction which the authors indicated should be mitomiRs.  

Response. We appreciate the reviewer’s suggestions. We have removed the unnecessary sentences/section from the introductory paragraphs and added the new available literature on mitomiRs and AD in the revised manuscript.

Comment 2. mitomiR is a general term for those miRNAs that either translocated into mitochondria or associated with/enriched in mitochondria. The location of mitomiR could have different mechanistic functions in the cell. Authors should clarify this important aspect of mitomiR when they discuss the relevant function of the mitomiR.  

Response. We appreciate the reviewer’s suggestion. We have included new literature on “MitomiRs and their cellular function” in relevance to mitochondria and other cellular functions in the revised manuscript (Section 8).

Comment 3. Mixed mitomiR and other miRNA concepts: per commonly agreed, mitomiRs are those miRNAs that reside in mitochondria (including translocated or associated/enriched in mitochondria) and function to regulate gene expressions including genes that are important to mitochondria function; other miRNAs are those that do not physically reside on mitochondria but regulate genes involve in mitochondria function. Following authors’ manuscript title and abstract, mitomiRs should be the focus of the discussion. Unfortunately, only a small portion of the manuscript content is relevant to these miRNAs. Many of mentioned miRNAs have no proven to be resided in mitochondria as ‘mitomiRs’. Authors should clarify if they categorized these miRNAs differently than the commonly agreed.

Response. The reviewer raised a very important aspect that was missing in the manuscript. We totally agreed with the reviewer. As suggested, we have added some new information onMitomiRs and regulation of mitochondrial function” and “Other cellular miRNAs and regulation of mitochondrial function” in the revised manuscript (Section 9).

Comment 4. Even though this is supposed to be a perspective article, the hypothesis/perspective still should be proposed based primarily on scientific findings. The authors intended to discuss the role of mitomiRs in AD/aging, however, no mitomiR was mentioned to be experimentally confirmed as CNS mitomiR in this manuscript. 

Response. We understand the reviewer’s concern. We did further literature search on CNS MitomiRs and we found very limited literature on this particular topic. We have added a new section on the “CNS associated MitomiRs and mitochondrial function” in the revised manuscript (Section 10).

Comment 5. The future direction is marked with several questions, but no actual perspective or direction is given.   

Response. We appreciate the reviewer’s concern. We added the possible future directions in the revised manuscript.

Comment 6. Line 150: “Based on these reports, it is confirmed that mitochondria retain local miRNAs synthesis mechanism, some potential miRNAs and their precursors.” This is an inaccurate statement. Argonaut protein complexes are mainly responsible for miRNA executive function. The existence of AGO complex in mitochondria suggests that miRNAs found in mitochondria may be functional but do not confirm that miRNAs are synthesized in mitochondria.

Response. We appreciate the reviewer’s careful checking. We have corrected this statement in the revised manuscript.

Response.

Comment 7. Line 149: ‘…and identified human miRNAs that we termed ‘mito-miRs’.’ Bendiera et al (2011) already named the subset of miRNA as ‘mitomiRs’. I don’t see any point to rename these miRNA as ‘mito-miRs’

Response. We appreciate the reviewer’s suggestions. We have removed this sentence in the revised manuscript.

Comment 8. The authors are encouraged to directly utilize original research articles in proposing putative function or mechanism of mitomiRs.

Response. We appreciate the reviewer’s suggestions. We have cited original references in the revised manuscript.

Comment 9. The following sentence is identical to a publication. The reviewer suggest the authors modify the sentence. Authors’ manuscript Line 147: ‘…provides the first comprehensive view of the localization of RNA interference components to the mitochondria’ Bendiera et al (2011) Abstract: “This study provides the first comprehensive view of the localization of RNA interference components to the mitochondria”

Response. Our sincere apologies for repeating the same sentence. We have removed this statement in the revised manuscript.

Comment 10. Figure 1 did not add any useful information to the manuscript.

Response. We understand the reviewer’s concern, so we removed Figure 1 and added a new Table 1, in the revised manuscript.

Comment 11. The manuscript is hard to read and largely unpolished. For example, line 84 ‘MiRNAs are the potential regulators of human genes’—miRNAs are proven regulators of human genes! the acronym ‘miRNAs’ or ‘miRNA’ already appear before the line 93 ‘MicroRNAs (miRNAs) are small..’  and miRNA, microRNA had been used randomly throughout the text. There are many more such minor editing issues in the manuscript.

Response. Our sincere apology for any grammatical errors. We have carefully checked the grammar and tried our best to remove all the errors.  

Reviewer 2 Report

In this article “Mitochondria localized microRNAs: An unexplored miRNA niche in Alzheimer’s disease and aging”, the authors reported the current insights and future research direction on mitochondrial miRNAs with focus on AD and aging. .

They further reported, a deeper understanding of mitochondrial miRNAs could help to develop mitochondrial biomarkers that could detect early stages of AD. Further, mitochondrial miRNAs research will help to understand mitochondria-based disease pathobiology and to develop the effective therapeutic strategies against AD.

The introduction, figures, Summary and future directions   and references are framed correctly. Overall, the article is informative for the scientific fraternity. I recommend the paper for publication after rectifying the below mentioned minor corrections.

Line No.29       :   is as a

Line No.146     :   co-immunoprecipitation studied of   

Line No.. 264  :   previous article

Line No.280     :  MiRNA

Author Response

Reviewer 2- Comments and Suggestions for Authors

In this article “Mitochondria localized microRNAs: An unexplored miRNA niche in Alzheimer’s disease and aging”, the authors reported the current insights and future research direction on mitochondrial miRNAs with focus on AD and aging.

They further reported, a deeper understanding of mitochondrial miRNAs could help to develop mitochondrial biomarkers that could detect early stages of AD. Further, mitochondrial miRNAs research will help to understand mitochondria-based disease pathobiology and to develop effective therapeutic strategies against AD.

The introduction, figures, Summary and future directions and references are framed correctly. Overall, the article is informative for the scientific fraternity. I recommend the paper for publication after rectifying the below mentioned minor corrections.

Response. We sincerely appreciate the reviewer’s positive comments and careful checking. We have corrected the following errors in the revised manuscript.

Line No.29       :   is as a

Line No.146     :   co-immunoprecipitation studied of    

Line No.. 264  :   previous article 

Line No.280     :  MiRNA

Reviewer 3 Report

The crosstalk exists among oxidative stress, mitochondrial dysfunction, and miRNA dysregulation plays a pivotal role in the onset and progression of neurodegenerative diseases such as AD. The current manuscript provides a novel perspective on mitochondrial miRNAs and their role and impact in AD. Following are major and minor comments. 

Major:

  1. All major results and discussions point back to the review published by the same authors (more than 10 times). Authors need to differentiate how the current manuscript differs from Gowda et al. 2021. Also, all results/discussions need to refer to original articles.

  2. Could you summarise the main incentive for this manuscript apart from its being interesting, maybe towards the end of Section 1? 

  3. Sections 6 and 7 elaborate on synaptic dysfunction in Alzheimer's disease with respect to mt-miRNA. Authors have largely concentrated on results from one article, ie. Gowda 2021. 

  4. Line 204-207: One of the thought experiments I would like to do is to extract information about miRNA and genes/proteins interaction and understand what top interactions come up. Did the authors already think about it?

  5. As I read, I understood that there is a handful of miRNAs known to play a role in AD. It would be a good idea to put this information in table/figure format. Also, little is known about miRNA-gene/protein interactions. This information can be extracted from public databases such as STRING.  I believe these two abstractions would greatly increase the impact of the current manuscript.

Minor:

  1. In many places, sentences start with “A study in YEAR.” It will be impactful if such a start is only used for recent literature/strong papers.

  2. Spell check: line 264

3. The words micro-RNA and miRNA are used inconsistently.

Author Response

Reviewer 3- Comments and Suggestions for Authors

The crosstalk exists among oxidative stress, mitochondrial dysfunction, and miRNA dysregulation plays a pivotal role in the onset and progression of neurodegenerative diseases such as AD. The current manuscript provides a novel perspective on mitochondrial miRNAs and their role and impact in AD. The following are major and minor comments. 

Major:

Comment 1. All major results and discussions point back to the review published by the same authors (more than 10 times). Authors need to differentiate how the current manuscript differs from Gowda et al. 2021. Also, all results/discussions need to refer to original articles.

Response. We understand the reviewer’s concern. We have added the original reference in revised wherever appropriate. We also explained how the current manuscript is different from our previous review article (Section 1).  

Comment 2. Could you summarize the main incentive for this manuscript apart from its being interesting, maybe towards the end of Section 1? 

Response. We understand the reviewer’s concern. We have added some lines at the end of section 1 to summarize the main incentive of the manuscript.

Comment 3. Sections 6 and 7 elaborate on synaptic dysfunction in Alzheimer's disease with respect to mt-miRNA. Authors have largely concentrated on results from one article, ie. Gowda 2021. 

Response. We understand the reviewer’s concern. We have revised sections 6 and 7 and added original references in the revised manuscript.

Comment 4. Line 204-207: One of the thought experiments I would like to do is to extract information about miRNA and genes/proteins interaction and understand what top interactions come up. Did the authors already think about it?

Response. We appreciate the reviewer’s suggestions. Yes, we proposed experiments for the same and currently working on the multi-Omics analysis of brain mitochondrial miRNAs in AD vs healthy mitochondria.

Comment 5. As I read, I understood that there is a handful of miRNAs known to play a role in AD. It would be a good idea to put this information in table/figure format. Also, little is known about miRNA-gene/protein interactions. This information can be extracted from public databases such as STRING.  I believe these two abstractions would greatly increase the impact of the current manuscript.

Response. We greatly appreciate the reviewer’s suggestion. We have added a new table to summarize the mitomiRs and their roles in AD and other diseases. We also appreciate the reviewer’s suggestion for extracting mitochondrial miRNAs-protein data from the publics database. However, we did not find the miRNAs and protein studied of mitochondrial genome on the same samples. This is a great idea, and we are planning to study the brain mitochondrial miRNAs and proteins within same samples in AD vs healthy controls.

Minor:

Comment. In many places, sentences start with “A study in YEAR.” It will be impactful if such a start is only used for recent literature/strong papers.

Response. We appreciate the reviewer’s suggestion. We have revised the sentences in the revised manuscript. 

Comment. Spell check: line 264

Response. Our sincere apology for any grammatical errors. We have carefully checked the grammar and tried our best to remove all the errors. 

Comment. The words micro-RNA and miRNA are used inconsistently.

Response. Our sincere apology for any grammatical errors. We followed the same pattern in the revised manuscript.  

Round 2

Reviewer 1 Report

The reviewer appreciated authors’ effort attempting to improve the manuscript. However, the authors fail to address my concerns sufficiently and the manuscript remains conceptually and structurally problematic. Thus, the reviewer is unable to recommend the current manuscript for publication. Some major issues are detailed below.

1. Again, mitomiR is a general term for those miRNAs that either translocated into mitochondria or associated with/enriched in mitochondria. Authors fail to clarify and continue to mis-present those miRNAs regulating mitochondria function from the conventional recognized mitomiRs.

Line 168: “Based on their roles and association with mitochondrial activities and functions, we categorized them as ‘mitochondrial miRNAs’.” This term is rather different from what they implied in the title “Mitochondria localized microRNAs…”. The authors need to give their so call ‘mitochondrial miRNAs’ a different name to avoid misperception of their mitochondrial function regulating miRNAs with conventional recognized mitomiRs.

2. The authors intended to discuss the role of mitomiRs in AD/aging, however, no mitomiR was mentioned to be experimentally confirmed as CNS mitomiR in this manuscript (previous comment). The added section “CNS associated MitomiRs and mitochondrial function” did not contain any CNS mitomiR. It is worth noting that brain tissue mitomiRs have been identified previously in both rodent and human brain tissues (see PMID: 25562527, PMID: 32451872).

3. The revised manuscript remained fairly unpolished. A couple examples:

In Table 1: 1) Column title ‘MitomiRs’-this is not accurate. Most of listed miRNAs are not mitomiRs; 2) Column title ‘Location’— ‘Mitochondria’: organelle level, ‘Cervical ganglion cells’: cellular level, and ‘Brain’: organ level. Authors need to be consistence in the level of a location!

Line 54-55: “The main incentive of the current article is to unveil the brain mitochondria localized/associated miRNAs and how they are important to maintain normal mitochondrial function. Furthermore, our study will shed light on the deregulated brain mitochondrial miRNAs and how they are involved in AD progression and pathogenesis”

The authors indicated they found no brain mitochondrial miRNAs in the literature—how they will “shed light on the deregulated brain mitochondrial miRNAs and how they are involved in AD progression and pathogenesis”

Line 126-129 “The functional relevance of miRNAs in mitochondria was further supported by the finding of Argonaute 2 localization to mitochondria and further the co-immunoprecipitation study of the mitochondrial transcript COX3 [14].” Line278 “Mitochondrial damage and cell stress beings promoting inflammation at neuronal synapses that spread across the membrane.” What do these sentences mean?

Author Response

Reviewer 1- Comments and Suggestions for Authors

The reviewer appreciated authors’ effort attempting to improve the manuscript. However, the authors fail to address my concerns sufficiently and the manuscript remains conceptually and structurally problematic. Thus, the reviewer is unable to recommend the current manuscript for publication. Some major issues are detailed below.

Response. Our sincere apology for not satisfying the reviewer with the revised version. In the second revision, we have addressed all the concerns that reviewer raised specially dissecting mitochondria localized miRNAs and mitochondria associated miRNAs.

Comment 11. Again, mitomiR is a general term for those miRNAs that either translocated into mitochondria or associated with/enriched in mitochondria. Authors fail to clarify and continue to mis-present those miRNAs regulating mitochondria function from the conventional recognized mitomiRs.  

Response. We appreciate the reviewer’s suggestions. In the revised version, we have dissected the mitochondria localized miRNAs and mitochondria associated miRNAs in Table 1. 

Comment 2. Line 168: “Based on their roles and association with mitochondrial activities and functions, we categorized them as ‘mitochondrial miRNAs’.” This term is rather different from what they implied in the title “Mitochondria localized microRNAs…”. The authors need to give their so call ‘mitochondrial miRNAs’ a different name to avoid misperception of their mitochondrial function regulating miRNAs with conventional recognized mitomiRs.  

Response. We appreciate the reviewer’s suggestion. We have revised the sentence in the revised manuscript.

Comment 3. The authors intended to discuss the role of mitomiRs in AD/aging, however, no mitomiR was mentioned to be experimentally confirmed as CNS mitomiR in this manuscript (previous comment). The added section “CNS associated MitomiRs and mitochondrial function” did not contain any CNS mitomiR. It is worth noting that brain tissue mitomiRs have been identified previously in both rodent and human brain tissues (see PMID: 25562527, PMID: 32451872).

Response. We appreciate the reviewer’s concern. As suggested, we have added new information and references in the CNS mitochondrial miRNAs section.

Comment 4. The revised manuscript remained fairly unpolished. A couple examples:

In Table 1: 1) Column title ‘MitomiRs’-this is not accurate. Most of listed miRNAs are not mitomiRs; 2) Column title ‘Location’— ‘Mitochondria’: organelle level, ‘Cervical ganglion cells’: cellular level, and ‘Brain’: organ level. Authors need to be consistence in the level of a location! 

Response. Our sincere apology for the confusion. We have revised the table as per reviewer suggestion.

Comment 5. Line 54-55: “The main incentive of the current article is to unveil the brain mitochondria localized/associated miRNAs and how they are important to maintain normal mitochondrial function. Furthermore, our study will shed light on the deregulated brain mitochondrial miRNAs and how they are involved in AD progression and pathogenesis”

The authors indicated they found no brain mitochondrial miRNAs in the literature—how they will “shed light on the deregulated brain mitochondrial miRNAs and how they are involved in AD progression and pathogenesis”.   

Response. We appreciate the reviewer’s concern. We have revised the sentence in the revised manuscript.

Comment 6. Line 126-129 “The functional relevance of miRNAs in mitochondria was further supported by the finding of Argonaute 2 localization to mitochondria and further the co-immunoprecipitation study of the mitochondrial transcript COX3 [14].” Line278 “Mitochondrial damage and cell stress beings promoting inflammation at neuronal synapses that spread across the membrane.” What do these sentences mean?.

Response. We appreciate the reviewer’s careful checking. We have corrected this statement in the revised manuscript.